# Colistin Resistance Mechanism and Management Strategies of Colistin-Resistant *Acinetobacter baumannii* Infections

**DOI:** 10.3390/pathogens13121049

**Published:** 2024-11-28

**Authors:** Md Minarul Islam, Da Eun Jung, Woo Shik Shin, Man Hwan Oh

**Affiliations:** 1Smart Animal Bio Institute, Dankook University, Cheonan 31116, Republic of Korea; mislambcmb@gmail.com; 2Department of Microbiology, College of Science and Technology, Dankook University, Cheonan 31116, Republic of Korea; jeongdaeun6052@naver.com; 3Department of Pharmaceutical Sciences, Northeast Ohio Medical University, Rootstown, OH 44272, USA; 4Center for Bio-Medical Engineering Core Facility, Dankook University, Cheonan 31116, Republic of Korea

**Keywords:** colistin, antibiotic-resistant, *A. baumannii*, mutation, lipid A

## Abstract

The emergence of antibiotic-resistant *Acinetobacter baumannii* (*A. baumannii*) is a pressing threat in clinical settings. Colistin is currently a widely used treatment for multidrug-resistant *A. baumannii*, serving as the last line of defense. However, reports of colistin-resistant strains of *A. baumannii* have emerged, underscoring the urgent need to develop alternative medications to combat these serious pathogens. To resist colistin, *A. baumannii* has developed several mechanisms. These include the loss of outer membrane lipopolysaccharides (LPSs) due to mutation of LPS biosynthetic genes, modification of lipid A (a constituent of LPSs) structure through the addition of phosphoethanolamine (PEtN) moieties to the lipid A component by overexpression of chromosomal pmrCAB operon genes and *eptA* gene, or acquisition of plasmid-encoded *mcr* genes through horizontal gene transfer. Other resistance mechanisms involve alterations of outer membrane permeability through porins, the expulsion of colistin by efflux pumps, and heteroresistance. In response to the rising threat of colistin-resistant *A. baumannii*, researchers have developed various treatment strategies, including antibiotic combination therapy, adjuvants to potentiate antibiotic activity, repurposing existing drugs, antimicrobial peptides, nanotechnology, photodynamic therapy, CRISPR/Cas, and phage therapy. While many of these strategies have shown promise in vitro and in vivo, further clinical trials are necessary to ensure their efficacy and widen their clinical applications. Ongoing research is essential for identifying the most effective therapeutic strategies to manage colistin-resistant *A. baumannii.* This review explores the genetic mechanisms underlying colistin resistance and assesses potential treatment options for this challenging pathogen.

## 1. Introduction

Multidrug-resistant *Acinetobacter baumannii (A. baumannii)* is a notorious pathogen, posing a great threat to human health, and is responsible for bacteremia, pneumonia, urinary tract, and skin and soft tissue infections, contributing to significant morbidity and mortality [1]. The World Health Organization (WHO) and the Centers for Disease Control and Prevention (CDC) have declared it a critical priority pathogen and urged the exploration of new treatment options to combat this pathogen [2,3]. The rise of multidrug-resistant (MDR) and extensively drug-resistant (XDR) strains of *A. baumannii* presents significant challenges in treating associated infections. Several antibiotics are typically considered effective for these infections, including polymyxin E and B, sulbactam, tigecycline, piperacillin/tazobactam, cefiderocol, and aminoglycosides. These antibiotics can be used alone or in combination to enhance treatment effectiveness [4,5]. Nevertheless, the resistant strains are becoming more rampant, resulting in limited treatment options [6].

Polymyxin antibiotics, mainly colistin, are currently used as a last-line defense against MDR *A. baumannii* infections. Colistin is a polycationic peptide that belongs to the class of polymyxin antibiotics, with only two members—polymyxin B and colistin—used in clinical settings [7]. It was first isolated in 1947 from the bacterium *Paenibacillus polymyxa subspecies Colistinus* and introduced into clinical use in the 1950s, but its use in human medicine was mainly limited to the treatment of pulmonary infections caused by MDR Gram-negative pathogens in patients with cystic fibrosis [8]. Colistin, initially limited to topical use in human medicine due to its nephrotoxic and neurotoxic risks when administered systemically, has increasingly become a last-resort antibiotic for challenging infections caused by multidrug-resistant (MDR) Gram-negative pathogens [9,10]. It is often used in critical situations, such as bacteremia, sepsis, and ventilator-associated pneumonia (VAP) in intensive care units. Additionally, colistin is utilized as an alternative treatment for urinary tract infections, osteomyelitis, joint infections, meningitis, pneumonia, gastrointestinal infections, pyoderma, soft tissue infections, as well as eye and ear infections [11]. Colistin is still frequently used as an additive in livestock feed to promote growth and treat intestinal infections. It is primarily used in food-producing animals such as pigs and poultry to manage these infections. Additionally, in the seafood industries, colistin sulfate is also used to promote fish growth [12,13]. However, overreliance on colistin in animals and human health exerts high selective pressure. 

Consequently, the growing clinical reliance on colistin for the treatment of infections has led to a surge of colistin resistance among clinical strains [14] (Table 1). Colistin resistance in *A. baumannii* can arise from several different mechanisms. One significant mechanism involves the addition of 4-amino-4-deoxy-L-arabinose (L-Ara4N) or phosphoethanolamine (pEtN) to the lipid A component of lipopolysaccharides (LPSs). This addition reduces the overall negative charge of the bacterial outer membrane (OM), making it more stable and hindering colistin penetration [15]. The PmrAB two-component system plays a role in regulating colistin resistance by controlling the expression of *PmrC*, an enzyme that facilitates the addition of pEtN to lipid A. This modification decreases the negative charge of the outer membrane, making it harder for colistin to bind [16]. Furthermore, mutations in genes responsible for LPS biosynthesis, such as *lpxC*, *lpxA*, or *lpxD*, can lead to the loss of LPSs and contribute to colistin resistance [17,18]. In addition, in *A. baumannii*, the plasmid carrying the mobile colistin resistance *(mcr)-1* gene contributes to colistin resistance by encoding a phosphoethanolamine transferase. This enzyme catalyzes the addition of a phosphoethanolamine group to lipid A in the bacterial outer membrane, potentially altering its structure [19]. Interestingly, many *A. baumannii* isolates have been frequently reported to exhibit heteroresistance to colistin. This phenomenon involves colistin-susceptible strains harboring a subpopulation of colistin-resistant cells. Under selective pressure, both in vitro and in vivo, heteroresistant *A. baumannii* strains can quickly evolve into strains with a high level of colistin resistance. Treating infections caused by heteroresistant isolates may lead to the selection of colistin-resistant subpopulations, potentially resulting in therapeutic failures [20].

In this review, we discuss the colistin-resistant mechanisms of *A. baumannii* and explore alternative therapeutic drugs to address colistin-resistant *A. baumanii* (Col-RAB) infections.

## 2. Colistin Mechanism of Action

Colistin exerts its antibacterial activity on the outer membrane, which harbors lipopolysaccharides that constitute the cell surface [36]. LPS molecules play a crucial role in limiting the entry of hydrophobic components and antibiotics, while also providing stability and integrity to the outer membrane of Gram-negative bacteria. Colistin, which is positively charged, binds to the negatively charged phosphate group of lipid A through electrostatic interactions. Lipid A is a hydrophobic component of LPSs and is significant for bacterial permeability and interaction with the external environment. Following the initial interaction, colistin competitively displaces divalent cations such as calcium (Ca^2+^) and magnesium (Mg^2+^), which destabilizes the cytoplasmic membrane. This disruption weakens the outer membrane’s LPSs and leads to the loss of inner cellular contents, ultimately resulting in bacterial cell death [37]. Colistin binds to lipid A (lipid A is an endotoxin) and exerts anti-endotoxin activity by neutralizing its effects. Thus, colistin inhibits the endotoxin functions of lipid A by binding to the LPS [38,39]. Colistin may promote the accumulation of reactive oxygen species (ROS) within cells, contributing to its bactericidal effect, particularly against Col-RAB. As colistin crosses the outer and inner membranes, the generated ROS are converted into hydrogen peroxide (H_2_O_2_) by superoxide dismutase enzymes. This H_2_O_2_ then participates in the Fenton reaction, which oxidizes Fe^2+^ to Fe^3+^ while simultaneously producing hydroxyl radicals (•OH). This reaction can lead to oxidative damage to both DNA and membrane lipids [40,41] (Figure 1).

## 3. Mechanism of Colistin Resistance in *A. baumannii*

*A. baumannii* can rapidly develop resistance to colistin through two main mechanisms, both of which involve either a complete loss of or significant alterations of the lipid A component of LPSs, the primary target of colistin activity. In both scenarios, the net negative charge on the cell surface decreases, affecting the electrostatic interaction between positively charged colistin and the negatively charged lipid A. The loss of LPSs is achieved through the inactivation of genes (*lpxA*, *lpxC*, and *lpxD* genes) involved in the initial step of lipid A synthesis. The most common mechanism, however, involves the modification of lipid A through the addition of phosphoethanolamine (PEtN). This modification is genetically driven by the chromosomal pmrCAB operon and *eptA* gene or through plasmid-borne *mcr* genes, all of which encode enzymes responsible for altering the structure of lipid A, thereby conferring resistance to colistin in *A. baumannii* (Figure 2).

### 3.1. Loss of Lipopolysaccharides Due to Mutations in Their Biosynthetic Genes

The genes *lpxA* and *lpxD* on the *A. baumannii* chromosome encode acyl-transferases involved in lipid A biosynthesis [42]. Similarly, *lpxB* and *lpxC*, also involved in lipid A biosynthesis, are located in separate chromosomal regions [43,44]. These genes primarily regulate the first three steps of the lipid A biosynthesis pathway and the hydrophobic anchor of lipopolysaccharides [45]. Therefore, disruption or mutation of *lpxA*, *lpxC*, and *lpxD* genes in *A. baumannii* may have accounted for the observed LPS deficiency leading to colistin resistance. Moffat and colleagues were the first to report that a lack of LPSs results in colistin resistance in *A. baumannii* [18]. They worked with 13 colistin-resistant variants of the *A. baumannii* strain ATCC 19606 and found mutations in the *lpxA*, *lpxC*, or *lpxD* genes involved in lipid A biosynthesis. These mutations led to a complete absence of LPS production. The absence of LPSs was confirmed in a clinical Col-RAB isolate. Carbohydrate silver staining of whole-cell lysates revealed the lack of LPSs in the colistin-resistant strains. Mutations in the *lpxA*, *lpxC*, and *lpxD* genes were identified through PCR amplification and sequencing, with mutations ranging from single nucleotide changes to large deletions up to 445 base pairs. These mutations affected critical residues in the lipid A biosynthesis pathway, such as P30 in *LpxC* and H159, G68, and Q72 in *LpxA*, highlighting their importance for the proper function of *LpxC* and *LpxA* [18]. Further research with Col-RAB strains has found that the insertion of *ISAba1* or *ISAba11* into the *lpxC* gene is a frequent occurrence, causing the inactivation of *lpxC* and *lpxA* genes. This inactivation leads to the loss of LPSs. Insertion sequence *ISAba11* was found to be inserted between nucleotides 390–393 in four strains and at nucleotide 420 or 421 in three strains, indicating that these specific regions may serve as hot spots for insertion [46]. In addition, under-expression of *lpxACD* has also been reported in colistin-resistant *A. baumannii*, leading to the decreased production of lipid A [47].

### 3.2. Modification of Lipopolysaccharides

The modification of lipid A moieties, such as the addition of molecules like 4-amino-4-deoxy-l-arabinose (L-Ara4N), phosphoethanolamine (PEtN), or galactosamine to lipid A, is another common mechanism of colistin resistance in *A. baumannii*. However, unlike other Gram-negative bacteria, *A. baumannii* lacks the biosynthesis machinery for L-Ara4N and instead utilizes PEtN or galactosamine to bind to lipid A [48,49]. The structural modification of lipid A reduces the net negative charge on the cell membrane. This change interferes with the binding of positively charged colistin to lipid A, thereby diminishing colistin’s effectiveness. Similar to enteric bacteria, the *PmrA/PmrB* two-component system (TCS) is involved in mediating colistin resistance in this bacterium [50,51]. The *pmrA/pmrB* system consists of a two-component response regulator (*PmrA*) and sensor kinase (*PmrB*) that enables bacteria to sense and respond to environmental cues. This system also regulates the *pmrC* gene, responsible for encoding a lipid A phosphoethanolamine (PEtN) transferase that alters lipid A in bacterial LPSs. Upon activation of the *PmrA/PmrB* system, phosphorylated *PmrA* interacts with the *pmrCAB* operon promoter, leading to increased expression and the synthesis of the enzyme responsible for adding PEtN to lipid A [16,52,53,54]. Colistin resistance develops when phosphoethanolamine (PEtN) is added to lipid A by the enzyme phosphoethanolamine transferase, known as *PmrC*, which is encoded by the *pmrC* gene. This modification process is regulated by the *PmrA/PmrB* two-component system (TCS). Mutations in the *pmrA*, *pmrB*, or *pmrC* genes can lead to the overexpression of *pmrC*, resulting in colistin resistance. Among these genes, *pmrB* is most commonly subjected to mutations, causing an increase in the expression of *pmrAB*. This overexpression enables *PmrC* to modify lipid A by adding PEtN, ultimately contributing to the development of colistin resistance [34,55]. A study identified two novel mutations in *pmrA* (I13M) and *pmrB* (Q270P) that contribute to colistin resistance. They also found a mutation in *miaA*, a posttranscriptional regulator previously reported to affect cell growth and virulence. The I221V mutation enhanced the colistin resistance of *pmrA* (P102R) [50]. Mutations in the *pmrB* gene of *A. baumannii*, which lead to the upregulation of *pmr* genes, are the most common cause of colistin resistance, with diverse amino acid changes in the *PmrB* protein frequently observed [54,56,57]. Nurtop et al. discovered the highest number of mutations in *pmrB* genes [58]. A study identified three colistin-resistant clinical *A. baumannii* strains from different patients. Non-synonymous mutations were present in various domains of *PmrB*, with a significant number concentrated in the HisKA domain, which is essential for autophosphorylation and transferring the phosphoryl group to the *PmrA* response regulator [57,59]. Mutations in the receiver domain of the response regulator *PmrA* were also identified in Col-RAB strains. Palethorpe et al. provided structural insights into the N-terminal domain of *A. baumannii PmrA* using X-ray crystallography and developed a full-length model through molecular modeling, which helped deduce the impact of two key *PmrA* mutations, PmrA::I13M and PmrA::P102R, both of which contribute to increased colistin resistance [60]. In addition, the *PmrAB* TCS regulates the *naxD* gene in colistin-resistant strains with activating mutations in *PmrB*. An activating mutation in *PmrB* can cause overexpression of *naxD*. *NaxD* deacetylates *N*-acetylgalactosamine to galactosamine, which is essential for incorporating galactosamine into lipid A. This modification adds a positive charge to lipid A, thereby enhancing resistance [61,62]. A study identified at least 20 amino acid substitutions in *PmrC* in colistin-susceptible isolates, which were associated with colistin MICs ranging from ≤0.125 µg/mL to 0.5 µg/mL [63]. In another study, the colistin-susceptible *A. baumannii* strain also adopts PEtN [49]. These findings indicate that the resistance mechanism may vary by strain, and the presence of PEtN alone may not completely explain colistin resistance. According to Beceiro et al., the emergence of moderate colistin resistance in *A. baumannii* is linked to distinct genetic alterations. These alterations comprise a minimum of one (01) point mutation in the *pmrB* gene, heightened expression of the *pmrAB* operon, and the activation of *pmrC*, leading to the incorporation of phosphoethanolamine into lipid A [64]

In addition, the overexpression of the *pmrC* homolog, named *eptA* (ethanolamine phosphotransferase A) is found in Col-RAB isolates. The mere presence of the *eptA* gene in bacteria does not inherently confer resistance. However, when the insertion sequences *ISAba1* integrate into the upstream of the *eptA* gene, they can enhance the expression of this enzyme, thus contributing to colistin resistance [34]. The gene *eptA* is frequently found in clinical strains of *A. baumannii* belonging to international clone II (IC2) and can exist in multiple copies within a single isolate (three or more). It is located distantly from the *pmrCAB* operon in the genome, which enhances resistance. For these strains to display colistin resistance, the integration of the *ISAbaI* element upstream of *eptA* is essential [34,35,65,66]. Lesho et al. discovered that colistin resistance was associated with point mutations in the *pmrA1* and/or *pmrB* genes. They also identified homologs of *pmrC*, named *eptA-1* and *eptA-2*, which were located distantly from the operon. Colistin-resistant isolates exhibited significantly increased expression of *pmrC1A1B*, *eptA-1*, and *eptA-2* compared to colistin-susceptible isolates [67]. A study of two *A. baumannii* isolates from a patient found increasing resistance to colistin before and after unsuccessful treatment. Genome sequencing identified an additional insertion sequence (*ISAba125*) in a transcriptional regulator gene associated with the highly resistant strain. Deletion of the *hns* gene in the less resistant strain increased its colistin resistance while restoring *hns* in the highly resistant strain reversed the resistance. Transcriptomic analysis revealed over 150 differentially expressed genes, including increased expression of *eptA*, linked to colistin resistance [68].

Plasmid-mediated transferable colistin resistance is mainly caused by mobile colistin resistance (*mcr*) genes, which produce phosphoethanolamine transferases (MCR enzymes). These enzymes alter lipid A of the LPS by adding phosphoethanolamine, a crucial mechanism of colistin resistance in Gram-negative bacteria [69,70,71]. To date, different versions of the *mcr-1* gene, from *mcr-1* to *mcr-10*, and their variants have been detected worldwide in various Gram-negative bacteria, particularly in food-producing animals [72,73,74,75,76]. In *A. baumannii*, two variants of *mcr*, *mcr-1* and *mcr-4.3*, have been reported. In 2019, *mcr-1* was first reported in clinical isolates from Pakistan [19]. Subsequently, it has been isolated from India [25], Egypt [23], China [77], and Iraq [78]. In 2008, *mcr-4.3* positive isolates of *A. baumannii* from Brazil were found to carry a novel plasmid (pAb-mcr4.3) containing *mcr-4.3* within a Tn3-like transposon [79]. The *mcr-4.3* gene was subsequently found in pig feces at a slaughterhouse in China, where it was carried by the plasmid pAB18PR065 through horizontal gene transfer [80]. It was also detected in isolates from the Czech Republic, hypothesizing that food imports can disseminate this gene in Europe [81].

### 3.3. Other Mechanisms Involved in Colistin Resistance

*A. baumannii* can develop resistance to colistin through mechanisms beyond the modifications or loss of LPS structure, one such as the involvement of efflux pumps. Lin et al. [82] proposed that *EmrAB* efflux pumps play a role in colistin resistance in *A. baumannii*. Studies have demonstrated that the overexpression of genes like *adeI*, *adeC*, *emrB*, *mexB*, and *macAB*, *HlyD* family, which encode efflux pump proteins, is associated with colistin resistance, as indicated by transcriptomic (RNAseq) analysis of resistant strains [83,84]. The contribution of efflux pumps is further supported by the reversal of resistance using efflux pump inhibitors (EPIs) and cyanide-3-chlorophenylhydrazone (CCCP) in various bacteria, including *A. baumannii*, *P. aeruginosa*, *K. pneumoniae*, and *S. maltophilia* [85,86]. Additionally, mutations in *vacJ*, *pldA*, *ttg2C*, *pheS*, and a conserved hypothetical protein have been linked to decreased colistin susceptibility through novel resistance mechanisms [87]. In a separate study, Paul et al. found notable changes in the transcription of genes encoding *sugE*, *ydhE*, *ydgE*, *mdfA*, *ynfA*, and *tolC* in Col-RAB when exposed to subinhibitory concentrations of colistin. This indicates that *A. baumannii* efficiently removes colistin using efflux pumps from the MATE and SMR families [88]. Outer membrane proteins (OMPs) are also implicated in the development of colistin resistance in *A. baumannii*. Specifically, OmpW plays a role in colistin binding and regulation of iron homeostasis in the bacterium [89]. The OMP inhibitor AOA-2 has been found to increase the susceptibility of both reference and clinical strains—both colistin-susceptible and colistin-resistant strains to colistin in vitro, highlighting the role of OMPs in colistin resistance. Furthermore, in vivo studies have shown that combining AOA-2 with colistin significantly reduces bacterial loads in tissues and blood and enhances mouse survival rates compared to colistin treatment alone [90]. Colistin resistance in *A. baumannii* may also be associated with non-*Lpx* proteins that support the structure and integrity of the outer membrane. One such protein, *LpsB*, is a glycosyltransferase essential for LPS core formation and contributes to the bacterium’s virulence. Deleting *LpsB* enhances susceptibility to colistin and cationic antimicrobial peptides, further emphasizing its role in maintaining outer membrane stability [91]. Heteroresistance is a progressive stage in the development of antimicrobial resistance, leading to the failure of anti-infective treatments [92]. It occurs when a small subset of bacterial cells in a genetically uniform group shows resistance to an antibiotic, while the majority of the population remains susceptible [93]. Li et al. first observed heteroresistance to colistin in 15 out of 16 clinical *A. baumannii* isolates. Subpopulations (<0.1%) of ATCC 19606 and most clinical isolates were able to grow in the presence of 3 to 10 μg/mL of colistin. Following four consecutive passages in colistin-containing broth (up to 200 μg/mL), the resistant subpopulation in ATCC 19606 increased from 0.000023% to 100%. Even after 16 passages in colistin-free broth, 2.1% of the population remained resistant [94].

## 4. Biological Cost of Acquiring Colistin-Resistant Trait

When bacteria become resistant to antibiotics due to genetic mutations, they often incur a "biological cost." This cost manifests as a decrease in overall fitness, pathogenicity, and virulence compared to non-resistant bacteria [95]. In vivo and in vitro studies have found lower fitness and reduced virulence potential in Col-RAB compared to their colistin-susceptible parental strains [96,97,98]. The LPS is considered a key factor in the virulence of *A. baumannii* [99]. As a crucial component of the outer membrane in Gram-negative bacteria, the LPS plays a vital role in creating a barrier that controls the passage of substances into the cell and helps maintain the structural integrity of bacteria [100]. Previous research has shown that the absence of LPSs in *A. baumannii* makes it more susceptible to antibiotics like azithromycin, rifampicin, and vancomycin [101]. A study investigated various virulence characteristics in colistin-resistant, LPS-deficient versions of the *A. baumannii ATCC 19606* strain and five multidrug-resistant clinical isolates and their LPS-deficient, colistin-resistant counterparts. The results indicated that the loss of LPSs caused growth deficiencies compared to the original strain both in laboratory conditions and in human serum. Additionally, the LPS-deficient strains showed reduced ability to grow and spread in live organisms. In a mouse model of widespread sepsis, infection with LPS-deficient strains resulted in lower levels of pro-inflammatory cytokines TNF-α and IL-6 in the bloodstream compared to infections with the parent strain, suggesting decreased virulence. The absence of LPSs also negatively impacted several virulence factors, including biofilm formation, surface motility, growth in low-iron environments, and increased susceptibility to various disinfectants used in healthcare settings. These findings highlight the significant impact of LPS loss on different virulence factors, shedding light on the relatively infrequent occurrence of LPS-deficient colistin-resistant strains in clinical practice [102]. Col-RAB strains that lack LPSs due to mutations in *lpx* genes only weakly activate neutrophils, leading to decreased production of reactive oxygen species and cytokines. Despite this, neutrophils exhibit a preference for killing these LPS-deficient strains, which are more susceptible to lysozyme and lactoferrin [103]. 

Mutations in the *pmrAB* two-component system result in colistin resistance in *A. baumannii*, which is correlated to impaired fitness, diminished virulence, lower biofilm formation, and decreased infectivity [104,105]. In one study, two strains of *A. baumannii* were isolated from a ventilator-associated pneumonia patient’s respiratory tract: One strain was colistin-susceptible (ABCS), while the other was colistin-resistant (ABCR). Whole-genome sequencing confirmed that ABCR evolved from parental strain due to mutations in the *pmrA* and *rpoB* genes and the absence of a prophage. The colistin resistance in ABCR was attributed to a *pmrA* mutation (E8D), indicating a close relationship between reduced virulence and colistin resistance. The loss of the prophage in ABCR may have also contributed to its decreased in vivo virulence [106]. In contrast, another Col-RAB isolate (CR17) remains sporadic, although colistin resistance was attained through *pmrAB* mutation [107]. While strain CR17 remained infectious, it showed decreased virulence and fitness in a murine sepsis model compared to the original susceptible strain, colistin-susceptible 01 (CS01). CS01 was more virulent than CR17, resulting in higher mortality rates and a shorter time to death [108]. Pournaras et al. discovered that the two colistin-resistant isolates (Ab249 and Ab347, with colistin MICs of 128 and 32 μg/mL, respectively) acquired resistance through *pmrAB* mutation and exhibited slower growth compared to their colistin-susceptible clinical counterparts, Ab248 and Ab299 (both having a colistin MIC of 0.5 μg/mL), indicating reduced fitness. Furthermore, one colistin-resistant isolate demonstrated reduced expression of the *Csu* system, which is crucial for biofilm formation, along with the outer membrane protein CarO and antioxidant proteins that defend against ROS generated by macrophages [41]. Further studies by the same researchers showed that *A. baumannii* strains, Ab249 and Ab347, had a notable decrease in biofilm formation in both static and dynamic assays (*p* < 0.001) [109]. Nonetheless, a study revealed that LPS-deficient (*lpxD* mutant) colistin-resistant strains exhibited a reduced growth rate, biofilm formation, and biofilm-associated gene expression compared to their colistin-susceptible counterparts. These strains also showed increased susceptibility to azithromycin, vancomycin, and rifampin. In contrast, LPS-modified (*pmrB* mutant) strains did not display significant differences in these characteristics. The loss of LPSs also hindered surface motility, while the expression of type IV pili remained unaffected [110]. LPSs play a critical role in biofilm formation in *A. baumannii*. The absence of LPSs can hurt biofilm development, as they are crucial for surface adhesion and structural stability. However, if LPSs undergo modifications instead of a complete loss, biofilm formation may still occur, as these modifications may not disrupt the essential processes involved in biofilm formation. LPS loss mutants incur higher fitness costs than *pmrB* mutants in *A. baumannii* [111].

The whole-genome sequencing (WGS) data of four clinical *A. baumannii* strains isolated from patients treated with colistin were analyzed to study the evolution and regression of resistance. Colistin resistance developed in all four patients due to mutations in the *pmr* locus. However, in three instances, susceptible strains became dominant after colistin treatment was discontinued. In one case, resistance was lost because of a compensatory mutation that reduced the chances of regaining resistance. Despite initial indications of fitness costs, genomic analysis revealed stable resistance that was not detected by standard clinical tests. Transcriptional studies confirmed increased *pmr* expression, and adjustments in environmental conditions restored the resistance phenotype [112]. In another study, an immunocompromised patient developed a Col-RAB infection while on long-term colistin therapy. The study examined phenotypic and genotypic characteristics, focusing on colistin resistance mechanisms and strain fitness. Both colistin-sensitive and colistin-resistant strains were extensively drug-resistant (XDR) and belonged to the same ST78 genotype. Unlike previous findings, the colistin resistance, attributed to a P233S mutation in the *PmrB* sensor kinase, did not lead to decreased fitness, growth, or virulence [113]. Moreover, whole-genome sequencing of five isolates showed that C440 and C428 were colistin-resistant and genetically identical to their susceptible counterparts. Colistin resistance in C440 was associated with a known P233S mutation in *pmrB*, while C428 had a novel ΔI19 mutation in *pmrB*. There were no discernible differences in virulence among isolates C080, C314, and C428 in vitro [114]. Further research is needed to explore the potential impact of compensatory mutations, post-translational modifications, or physiological alterations.

## 5. Available Therapies and Prospects for Col-RAB Infections

The increase in Col-RAB is a major global concern due to the rise of multidrug-resistant strains resulting from the overuse of colistin. This worrisome development threatens to regress clinicians and patients to a time with limited treatment options akin to the pre-antibiotic era [115]. Therefore, there is an urgent need for innovative therapeutic approaches to address *A. baumannii* infections. The management of Col-RAB is similar to that of carbapenem-resistant strains, but colistin alone is not recommended as a standalone treatment and may no longer be the primary component of combination therapy [116,117,118]. To tackle this issue, potential solutions include the development of new drugs, repurposing existing medications, and exploring combination therapies involving colistin with other agents or adjuvants. Promising strategies such as nanotechnology-based therapies, antimicrobial peptides (AMP), photodynamic therapy, vaccines, CRISPR interference (CRISPRi), and phage therapy offer hope for effectively combating Col-RAB.

### 5.1. Combination Therapy

The combination of colistin with other drugs has been proposed as an effective strategy to fight against MDR *A. baumannii* persister cells, which are known as combination therapy. When tobramycin is combined with colistin or ciprofloxacin, it can effectively eliminate persister cells of *A. baumannii* during the late exponential and stationary growth phases. Colistin helps tobramycin enter the cells by boosting membrane permeability and inducing hyperpolarization of the inner membrane, leading to higher levels of ROS production [119]. Other antibiotics, including econazole, tigecycline, meropenem, rifampin, fosfomycin, amikacin, ampicillin/sulbactam, polymyxin B, minocycline, eravacycline, ceftazidime/avibactam, trimethoprim/sulfamethoxazole, rifabutin, and azithromycin are promising combinations with colistin to combat Col-RAB [120,121,122,123,124,125,126,127,128,129,130,131]. In a study on colistin-based combinations, the synergistic effects of two-drug combinations using eight commercially available antibiotics were assessed using the checkerboard approach. Among the combinations tested, vancomycin, aztreonam, ceftazidime, and imipenem showed the highest potency, demonstrating synergistic activity against the strains studied [30]. The combination of fosfomycin or fusidic acid with colistin resulted in a significantly improved microbiological response, along with a potential enhancement in clinical outcomes and reduced mortality rates compared to colistin use alone. Fusidic acid was also effective in preventing the development of colistin resistance, a phenomenon that occurred more frequently when colistin was used as a standalone treatment [132,133]. Pregnadiene-11-hydroxy-16,17-epoxy-3,20-dione-1 (PYED-1), a steroid, has shown notable antimicrobial, antibiofilm, and antivirulence properties against both Gram-positive and Gram-negative bacteria, as well as *Candida spp*. When combined with colistin, PYED-1 can synergistically enhance its antimicrobial efficacy against multidrug-resistant strains of *A. baumannii* [134].

### 5.2. Adjuvant Treatment

Adjuvants are non-antibiotic molecules that boost the effectiveness of antibiotics such as colistin. They help lower the necessary treatment dose and reduce its harmful effects. These compounds function by enhancing the antibiotic’s penetration into bacterial cells or blocking its expulsion, thus restoring susceptibility in extensively drug-resistant (XDR) strains. This method can be applied to address infections resistant to conventional therapies, presenting a hopeful tactic to tackle drug-resistant bacteria by enhancing the efficiency of current antibiotics. Table 2 lists successful adjuvants used with colistin against Col-RAB.

### 5.3. Drug Repurposing

Drug repurposing is a potential approach in drug discovery that identifies new therapeutic uses for existing drugs to treat resistant infections. This strategy enhances the discovery process by leveraging the known pharmacological properties of these drugs, leading to faster and more cost-effective development compared to developing new drugs. By skipping preclinical trials and moving directly to Phase 2 to test drug efficacy, this approach offers economic advantages and rapidly expands the range of available infection treatments [145,146,147]. Drugs used for anthelmintic purposes, such as Niclosamide, closantel, and oxyclozanide, restore the colistin activity against Col-RAB and other Gram-negative bacteria when used in combination with colistin [148,149,150,151]. Tavaborole was approved by the FDA in July 2014 as an antifungal agent. A recent study by Borges et al. showed that tavaborole has strong antimicrobial activity, with a minimum inhibitory concentration (MIC) value of 2 μg/mL. It also demonstrated potent activity against both the standard ATCC strain and multidrug-resistant (MDR) clinical strains of *A. baumannii*, as well as effective efficacy against biofilms from multidrug-resistant strains at a concentration of 16 μg/mL [152]. Mitomycin C and 5-fluorouracil are anticancer agents, fluspirilene is an antipsychotic drug, and Bay 11-7082, an inhibitor of κB kinase, is a broad-spectrum drug that is effective against MDR *A. baumannii* [153,154]. In experimental pneumonia models in mice, the efficacy of *N*-desmethyltamoxifen, a metabolite of the anticancer agent tamoxifen, was assessed in combination with colistimethate sodium (CMS) or tigecycline. The results demonstrated significant effectiveness against *A. baumannii* and *E. coli* strains. Furthermore, combining *N*-desmethyltamoxifen with antibiotics led to a reduction in bacterial concentrations in the lungs and blood for both *A. baumannii* strains [155]. A study has identified *A. baumannii* dihydroorotate dehydrogenase (DHODH) as a potential target for treating drug-resistant infections. Researchers repurposed compounds from a malaria DHODH program and discovered potent inhibitors, such as DSM186, that exhibited strong activity against various *A. baumannii* strains. Another compound, DSM161, demonstrated protective effects in mouse infection models without any observed resistance. The structure of AbDHODH bound to DSM186 was resolved, confirming its potential as a target for developing new antimicrobial agents [156].

### 5.4. Antimicrobial Peptides

Antimicrobial peptides (AMPs) are an important substitute for antibiotics that affect a wide range of microbes. AMPs offer a potential avenue to fight against MDR, XDR, and Col-RAB through a variety of mechanisms [157,158]. Numerous studies on AMPs suggest that AMPs, either alone or in combination with colistin, show effectiveness against MDR, XDR, and Col-RAB. Peptides such as Esc (1-21), melittin, indolicidin, mastoparan, Ω76, NuriPep 1653, Cec4, 2K4L, LS-sarcotoxin, and LS-stomoxyn have demonstrated effectiveness against *A. baumannii* [159,160,161,162,163,164,165].

### 5.5. Nanoparticles (NPs)

Nanotechnology offers a promising approach to combat drug resistance by providing new treatment options and helping to conserve our antibiotic resources [166,167]. The antibacterial mechanisms of nanoparticles (NPs) are not yet fully understood. However, several well-recognized actions include disrupting bacterial membranes, generating ROS, depleting ATP, and inhibiting DNA synthesis [168]. Unlike antibiotics, NPs suppress microbes using multiple mechanisms that can act simultaneously. These multiple simultaneous mechanisms would necessitate several concurrent genetic mutations within a single bacterial cell to develop resistance, making it difficult for bacteria to develop resistance to NPs [169]. The antibacterial effect of NPs starts with their interaction with the bacterial surface, where they adhere to the cell wall. This adherence changes the membrane potential and increases permeability, allowing NPs to enter the cell and affect its structure and function. Once inside, NPs interact with key cellular components like DNA, ribosomes, lysosomes, and enzymes, causing oxidative stress, membrane disruption, electrolyte imbalances, enzyme inhibition, and alteration in gene expression [170,171]. ROS, such as hydrogen peroxide and superoxide anion, play a key role in damaging bacterial proteins, lipids, and DNA, contributing to bacterial eradication [172].

In *A. baumannii*, NPs help prevent biofilm formation by inhibiting the expression of genes associated with this process. Table 3 provides a summary of recent applications of nanoparticles in combating MDR *A. baumannii* strains. Silver nanoparticles (AgNPs) at a concentration of 25 µg/mL have been shown to reduce the expression of several virulence-related genes (*kpsMII* and *afa/draBC*) as well as biofilm-related genes (*bap*, *OmpA*, and *csuA/B*). This contributes to the mechanisms by which silver disrupts microbial growth [173]. Additionally, Gans and zinc oxide nanoparticles (ZnO NPs) downregulate the expression of efflux pump genes *adeA*, *adeC*, and *abeM*, which are involved in biofilm formation, leading to impaired biofilm development [174]. Nanoparticles in combination with colistin exert synergistic effects against Col-RAB. For example, the combination of colistin and silver nanoparticles significantly reduced bacterial growth and viability by inducing cellular damage, such as wrinkling, deformation, and cracks in the outer surface; this, however, was increased in colistin monotherapy [175,176,177]. Moreover, colistin-silver nanoparticle synergism led to a reduction in the MICs of colistin by more than four times [178]. In addition to AgNPs, a hexahistidine-tagged antimicrobial peptide (Lys AB2 P3-His) loaded onto DNA aptamer-functionalized gold nanoparticles (AuNP-Apt) significantly reduced *A. baumannii* colonization and increased survival in infected mice [179]. A study by Usjak et al. [180] revealed that a combination of low concentrations of colistin (0.5 or 1 μg/mL) and selenium nanoparticles (0.5 μg/mL) significantly decreased the initial bacterial load within the first 4 h of incubation. This effect was not observed with colistin alone at concentrations of 0.5, 1, or 2 μg/mL. The synergistic action of these agents at low doses implies that these concentrations could be feasibly achieved through systemic administration in vivo.

### 5.6. Photodynamic Therapy

Antimicrobial photodynamic therapy (APDT) has emerged as a cutting-edge technique in modern healthcare systems, offering a promising alternative to traditional antibiotic therapy for photo-inactivating a wide range of bacterial pathogens [196]. Boluki et al. evaluated photodynamic therapy (PDT) using toluidine blue O (TBO) and a light-emitting diode (LED) as a photosensitizer and radiation source on pan drug-resistant *A. baumannii* isolated from a burn patient. The study demonstrated that PDT was effective in eradicating bacterial infections regardless of resistance by directly modulating the activity of the *pmrA/pmrB* two-component regulatory system [197]. A subsequent study revealed that after exposing PDT (TBO and LED), the outer membrane protein A degrades due to the overexpression of the *ompA* gene in Col-RAB, which could assist in promoting antibiotic penetration [198]. In addition, PDT resulted in an 83.7% decrease in bacterial count compared to the control group. When PDT was used in combination with colistin, a robust synergistic effect was observed against *A. baumannii*, achieving a 100% kill rate and a nine-log reduction in bacterial load at all tested colistin concentrations. In addition, PDT significantly lowered colistin’s minimal inhibitory concentration (MIC) against Col-RAB by over 11-fold [199].

### 5.7. CRISPR Technology

CRISPR-associated gene editing could offer a new solution to combat PDR *A. baumannii*. Research has revealed that the clinical isolate AB43, equipped with a full I-Fb CRISPR-Cas system, utilizes the Cas3 nuclease to modulate quorum sensing and influence drug resistance. Through the downregulation of the quorum-sensing synthase *AbaI*, CRISPR-Cas activity resulted in decreased efflux pump activity, reduced biofilm formation, heightened ROS production, and diminished antibiotic resistance [200]. A study by Wang et al. utilized CRISPR/Cas9 to remove plasmids in *E. coli* isolates, making them more susceptible to antibiotics. Plasmids were successfully removed from isolate 14EC033, and multiple plasmids were eliminated in 14EC007. A single sgRNA deleted the colistin-resistant *mcr-1* gene in one plasmid, but unintended recombination occurred due to the presence of *IS5* upstream of *mcr-1* in p14EC033a. While the method was effective for plasmid elimination and gene deletion, caution should be taken to avoid unintended genetic alterations [201].

### 5.8. Bacteriophage Therapy

Phage therapy is one of the most potent therapeutic approaches gaining momentum to address Col-RAB infections. A lysin from bacteriophage PMK34 in combination with colistin resulted in up to a 32-fold reduction of the MIC of colistin and reverted colistin-resistant strains to susceptible in both Mueller–Hinton broth and 50% human serum [202]. The phage effectively reduced biofilms and prevented new ones. Additionally, a combination of the phage vWU2001 and colistin demonstrated decreased bacterial growth. In *Galleria mellonella*, the combined therapy improved survival and bacterial clearance compared to individual treatments, suggesting a synergistic effect against CRAB [203]. Another lytic phage, IsfAB78, isolated from wastewater was able to reduce the biofilm of Col-RAB by up to 87% [204]. In a clinical case, a 68-year-old diabetic patient with necrotizing pancreatitis developed a multidrug-resistant *A. baumannii* infection that was unresponsive to antibiotics, including colistin and tigecycline. After 4 months of worsening condition, bacteriophage therapy began using nine phages that targeted the bacteria. The treatment was administered intravenously and into abscesses, reversing the patient’s clinical deterioration and clearance of the infection. Minocycline was introduced five days into bacteriophage therapy, showing an additional effect in vitro against phage-resistant *A. baumannii*. This case highlights the potential of combining bacteriophages with antibiotics like colistin and minocycline to treat resistant infections [205].

## 6. Concluding Remarks

Taken together, Col-RAB poses a significant challenge to healthcare systems. To address this threat effectively, a comprehensive alternative strategy is necessary. Col-RAB strains often incur a fitness cost, resulting in reduced virulence due to growth defects and impaired infection capabilities. Understanding these fitness costs could provide insights into managing antibiotic resistance by exploiting the increased susceptibility to other antibiotics through combination therapies [206,207]. Although no single treatment has shown clear superiority, there are various non-polymyxin-based regimens available, with colistin still playing a role in synergistic combination therapies. Moreover, adjuvants, repurposing of existing drugs, AMPs, nanotechnology, photodynamic therapy, CRISPR/Cas, and phage therapy may as alternative treatment options in the future. 

## Figures and Tables

**Figure 1 pathogens-13-01049-f001:**
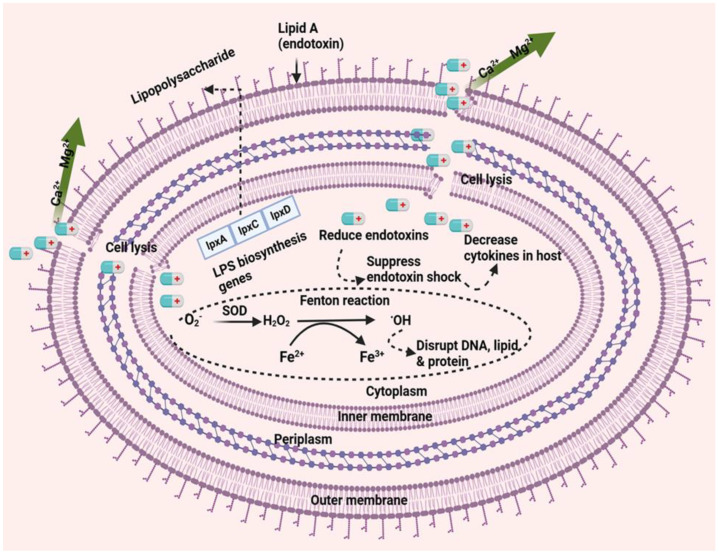
Mechanisms of action of colistin against *A. baumannii*. Positively charged colistin interacts with the negatively charged phosphate groups of lipid A, displacing Ca^2+^ and Mg^2+^ ions, leading to membrane disruption [37]. By binding to LPS, colistin neutralizes LPS endotoxin activity [39]. Additionally, colistin induces intracellular ROS accumulation, which is converted to H_2_O_2_ by superoxide dismutase. H_2_O_2_ participates in the Fenton reaction, oxidizing Fe^2+^ to Fe^3+^ and generating hydroxyl radicals (•OH) that disrupt iron homeostasis, causing oxidative damage to DNA and membrane lipids [40] (created with Biorender.com accessed at 29 October 2024). + represent the positive charge of colistin.

**Figure 2 pathogens-13-01049-f002:**
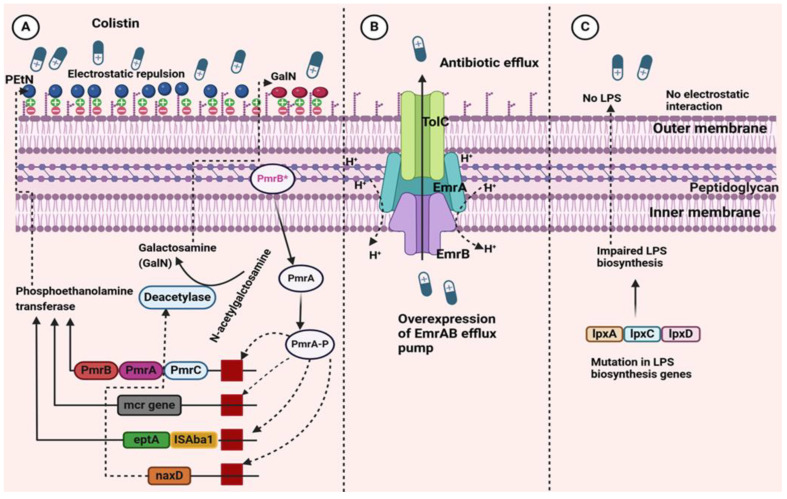
A schematic illustration showing the general mechanisms of colistin resistance in *A. baumannii.* (**A**) Colistin resistance occurs due to modification in lipid A of lipopolysaccharides (LPSs), reducing its negative charge and interfering with colistin binding. The *PmrA/PmrB* system controls the *pmrC* gene, which encodes phosphoethanolamine transferase adding phosphoethanolamine (PEtN) to lipid A. Plasmid-borne *mcr* genes also produce enzymes with this function. Mutations in *PmrB* can activate the *naxD* gene, modifying lipid A by adding galactosamine (GalN). Overexpression of *eptA*, often triggered by insertion sequences like *ISAba1*, also contributes to resistance. (**B**) Overexpression of the *emrAB* efflux pump expels colistin from the membrane. (**C**) Mutations or disruptions in lipid A biosynthesis genes *lpxA*, *lpxC*, or *lpxD*, cause LPS deficiency, leading to resistance (created with Biorender.com accessed at 22 November 2024).PmrB* means activated pmrB which phosphorylates the pmrA and increase the expression of pmrC.

**Table 1 pathogens-13-01049-t001:** Recent studies revealed different aspects of Col-RAB.

Col-RAB Isolated from	Period	Region	Study Report
ICU patients	June 2021 and May 2022 (published in 2024)	Jordan	20.6% of the *A. baumannii* clinical isolates were colistin-resistant [21].
ICU patients	January 2016 to October 2020 (published in 2022)	Iran	24% of the isolates exhibited resistance to colistin due to mutations in the *pmrB* and *lpxACD* genes [22].
Different clinical specimens	January 2020 to March 2021 (published in 2022)	Egypt	First to report the presence of the plasmid-mediated colistin-resistant *mcr-1* gene in a human clinical *A. baumannii* isolates in Egypt. In addition, colistin resistance was possibly due to pmrCAB overexpression or mutations [23].
Clinical and environmental samples	2016 to 2018 (published in 2019)	Iraq	Out of 121 *A. baumannii* isolates, 76% were resistant to colistin. The *mcr-1*, *mcr-2*, and *mcr-3* genes were found in 73.5%, 64.5%, and 67.8% of the isolates, respectively. Integrons containing intI2 were present in 64.5% of the isolates, intI3 in 66.9%, and the quorum-sensing *lasI* gene in 49.6% [24].
Hospitalized patient samples	Published in 2020	India	20% of clinical isolates are colistin-resistant, each carrying the *mcr-1* gene. This is the first reported case of *mcr-1* in a human clinical isolate of *A. baumannii* in India [25].
Hospital samples	2017 to 2019 (published in 2022)	Thailand	Colistin resistance in *A. baumannii* was 15.14%. No *mcr* gene was detected, and resistance was linked to lipid modification. Colistin and sulbactam remained effective treatments [26].
Clinical samples of patients	2019–2020 (published in 2022)	Pakistan	Colistin resistance in *A. baumannii* was overcome by the synergistic effect of colistin and doxycycline, along with the anti-inflammatory properties of *S. lappa* extract in infected mice [27].
ICU patients	June 2017 to October 2019 (published in 2020)	Greece	Critically ill patients with Col-RAB bloodstream infection face rapid mortality due to fulminant septic shock compared to those with colistin-susceptible *A. baumannii* infection [28].
Hospitalized patients	2018 and 2021 (published in 2023)	Serbia	Col-RAB isolates, mainly international clone II, exhibit increased *pmrC* expression due to *PmrAB* mutations, leading to the widespread distribution of high-risk clonal lineages [29].
Patients sample	2015 and 2016 (published in 2022)	Saudi Arabia	Vancomycin, aztreonam, ceftazidime, and imipenem exhibited synergistic activities against Col-RAB strains [30].
Patients with ventilator-associated pneumonia	2012 and 2015 (Published in 2020)	Greece, Italy, and Spain	Amino acid substitutions in PmrCAB are associated with increased *pmrC* expression and elevated colistin MICs [31].
Hospitalized patients	2016 (published in 2020)	Thailand	High abundance (44%) of colistin heteroresistance among the CRAB clinical isolates [32].
ICU patients	2020	Korea	Analysis of Col-RAB strains identified a core extracellular proteome of 506 proteins, including induced β-lactamases (*ADC-30*, *OXA-23*), porins (CarO, CarO-like), and other transport and receptor proteins. CarO-like porin was identified as an *A. baumannii*-specific marker through peptide analysis [33].
Clinical isolates	2019	Switzerland	A novel colistin resistance mechanism was identified in *A. baumannii* involving *ISAbaI*-mediated overexpression of the *PmrC* homolog *EptA* [34].
Hospitalized patients	2015 and 2017 (published in 2020)	Greece	Convergent evolution in Col-RAB isolates is driven by shared lipid A modifications and widespread resistance genes [35].

**Table 2 pathogens-13-01049-t002:** Compounds combine with colistin as adjuvants to fight against Col-RAB infections.

Compounds	Subjects	Mechanisms	References
Kaempferol	*Galleria mellonella*	Disruption of iron homeostasis and Fenton reaction lead to the generation of ROS.	[40]
Nitazoxanide	*Galleria mellonella*	Inhibit biofilm	[135]
2-aminoimidazole	*Galleria mellonella*		[136]
Chrysin	*Galleria mellonella*/Mouse	Damage extracellular membrane and modify bacterial membrane potential	[137]
Curcumin	In vitro	Increase ROS and function as efflux pump inhibitor	[138]
AOA-2	Mouse	OmpA inhibitor significantly decreases the bacterial cell load in tissues and blood and increases mouse survival rate	[90]
Panduratin A	In vitro	Inhibit biofilm formation	[139]
IMD-0354	Mouse	Inhibit lipid A modification	[140]
NMD-27	Mouse		[140]
Auranofin	Mouse	Inhibition of MCR	[141]
Essential oils of clove and thyme	In vitro	Increase cell membrane permeability	[142]
Disulfiram	Mouse	Damage bacterial membranes and disrupt metabolism	[143]
Capsaicin	Mouse	Inhibit the biofilm formation, disrupt outer membrane, and inhibition of protein synthesis and efflux pumps activity	[144]

**Table 3 pathogens-13-01049-t003:** Recent advances in nanoparticle research against MDR *A. baumannii*.

Nanoparticles	Effects on *A. baumannii*
AgNPs	Silver nanoparticles (AgNPs) can both trigger apoptosis and suppress new DNA synthesis in multidrug-resistant *A. baumannii* in a manner dependent on their concentration [181].
ZnO-NPs	ZnO-NPs combined with ciprofloxacin and cefotaxime effectively restored their antibacterial activity against MDR Acinetobacter. Additionally, ZnO-NPs improved colistin’s ability to inhibit growth and reduced the necessary dosage against MDR Acinetobacter [182].
Meropenem-loaded chitosan nanoparticles (MP-CS)	MP-CS nanoparticles matched free meropenem in fighting *A. baumannii* with lower drug levels, preventing cell infection and showing high biocompatibility [183].
Gallate-polyvinylpyrrolidone-capped hybrid silver nanoparticles (G-PVP–AgNPs)	G-PVP–AgNPs were found to have effective antimicrobial activity against carbapenem-resistant *A. baumannii* RS-307 through an ROS-dependent killing mechanism [184].
Nisin-Silver nanoparticles conjugate	Nisin’s antimicrobial activity is boosted when conjugated with AgNPs even at higher concentrations [185].
Selenium nanoparticle (SeNPs)	Inhibit biofilm formation of MDR *A baumannii* clinical isolates [186]
Chitosan-coated human albumin nanoparticles (haNPs)	haNPs combined with colistin significantly decreased MIC values, showing an inhibitory effect on biofilm formation that was 60-fold greater for colistin-resistant *A. baumannii* than for free colistin [187].
Copper oxide nanoparticles (CuO)	Copper oxide (CuO) nanoparticles show promise as an anti-capsular agent against *A. baumannii*, and their antimicrobial activity is enhanced when combined with gentamicin [188].
Biogenic silver nanoparticle (Bio-AgNP)	In combination with polymyxin B, Bio-AgNPs reduced the MIC of polymyxin B by 16-fold [189].
Aβ11/T80@CSs	Aβ11/T80@CSs nanoparticles successfully crossed the blood–brain barrier, delivering tigecycline to the cerebrospinal fluid and showing significant therapeutic potential against MDR *A. baumannii*-induced intracranial infections [190].
Poly ε-caprolactone (PCL) nanoparticles	Polymeric nanoparticles of ciprofloxacin and levofloxacin effectively reduce MICs and biofilms of MDR *A. baumannii* [191].
Arsenic nanoparticles (ARn)	Biosynthesized arsenic nanoparticles (ARn) using pink rose petals demonstrated significant antibacterial activity against MDR *A. baumannii* [192].
Lipoic acid-capped silver nanoparticles	Alginate-based aerogels with lipoic acid-capped AgNPs showed substantial antimicrobial, antioxidant, anti-inflammatory, and hemocompatibility properties against *A. baumannii* compared to commercial dressings [193].
Aluminium oxide nanoparticles (Al_2_O_3_ NPs)	Effectively inhibit biofilm formation, adhesion, and EPS production in MDR *A. baumannii*, showing low toxicity in HeLa cells [194].
Silica nanoparticles	Combined with low-dose amoxicillin effectively inhibits biofilm formation in *A. baumannii*, reducing toxicity compared to high doses [195].

## Data Availability

Not applicable.

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
