# Peer review of "Colistin Resistance Mechanism and Management Strategies of Colistin-Resistant Acinetobacter baumannii Infections"

_pathogens, 2024, doi:10.3390/pathogens13121049_

Round 1
Reviewer 1 Report
Comments and Suggestions for Authors
The following major corrections were requested for the review before being accepted for publication:
1- Provide a table summarizing recent studies reporting the resistance of colistin resistance among clinical isolates of A. baumannii
2- Nanoparticles of different types were reported to have potential antimicrobial activity against A. baumanni strains, add the recent studies discussing that in a separate subsection
3- Provide complete detailed description of mechanism of action of different types of nanomaterials against A. baumanni strains as reported previously
4- Many previous reports discussed the synergistic antibacterial efficiency of nanoparticles as silver or zinc oxide with colistin against drug resistant bacterial pathogens (add recent studies reporting that summarized in a table), the following reports could be added.
https://doi.org/10.1016/j.jiph.2021.09.015, https://doi.org/10.3390/cryst12081057, https://doi.org/10.1016/j.jksus.2024.103461, https://doi.org/10.1186/s12879-024-09471-1, https://doi.org/10.1166/jbn.2016.2273
Comments on the Quality of English Language
Moderate English corrections were required
Author Response
The following major corrections were requested for the review before being accepted for publication:
Dear reviewer,
Thank you for your valuable feedback on our manuscript, " Colistin Resistance Mechanism and Management Strategies of Colistin-Resistant Acinetobacter baumannii Infections." We have carefully reviewed and incorporated your suggestions. Your insightful comments have greatly improved our research, and we sincerely appreciate the time and effort you dedicated to the review process. Your constructive criticism will enhance the quality of our work, and we are grateful for your input. Thank you for your thorough review. Below, we address the comments you provided.
1-Provide a table summarizing recent studies reporting the resistance of colistin resistance among clinical isolates of A. baumannii
Reply: Thank you for this suggestion. We added a table that summarizes the recent findings on colistin-resistant A. baumannii.
2- Nanoparticles of different types were reported to have potential antimicrobial activity against A. baumanni strains, add the recent studies discussing that in a separate subsection
Reply: Thank you so much for your valuable suggestion. In the revised manuscript we discussed the therapeutic potency of nanoparticles in a separate subsection.
3-Provide a complete detailed description of the mechanism of action of different types of nanomaterials against A. baumanni strains as reported previously
Reply: Thank you so much. In the revised manuscript, we explained the detailed mechanism of action of nanoparticles in prokaryotic cells.
4-Many previous reports discussed the synergistic antibacterial efficiency of nanoparticles such as silver or zinc oxide with colistin against drug-resistant bacterial pathogens (add recent studies reporting that summarized in a table), the following reports could be added.
Reply: Thank you for providing us with recent articles on nanoparticles used in fighting pathogenic bacteria. These articles were very helpful and expanded our knowledge of the application of nanoparticles for therapeutic purposes. We tried to include as many recent studies as possible.
https://doi.org/10.1016/j.jiph.2021.09.015, https://doi.org/10.3390/cryst12081057, https://doi.org/10.1016/j.jksus.2024.103461, https://doi.org/10.1186/s12879-024-09471-1, https://doi.org/10.1166/jbn.2016.2273
Comments on the Quality of English Language- Moderate English corrections were required
Reply: Thank you for this suggestion. We thoroughly check the text and amend it where necessary.
Reviewer 2 Report
Comments and Suggestions for Authors
Thank you for letting me review this review on Colistin Resistance Mechanism and Management Strategies of Colistin-Resistant Acinetobacter baumannii Infections.
I think the authors have produced a very well written and carefully explained review. I was particularly impressed that the authors followed up their review by outlining current research into combating this resistance.
I have very little to add to these comments except some very minor typos and observations on the text.
I recommend that this publication is accepted for publication by the journal after some very minor typo corrections.
Minor comments
Line 19. Perhaps replace the words Bugs in the abstract with something more formal since it is an in-depth review.
Line 37. Should A. baumanii not be in italics ? Be careful of the spacing when you abbreviate.
Line 96. Confer?
Line 106. Is this reference the same as line 111?
Figure 2. Nice diagram.
Line 174. Not sure about starting a sentence with an abbreviated gene name?
Line 509. The authors are correct, extreme caution should be exercised using CRISPR methods to avoid unintended genetic alterations.
Author Response
Thank you for letting me review this review on Colistin Resistance Mechanism and Management Strategies of Colistin-Resistant Acinetobacter baumannii Infections.
I think the authors have produced a very well-written and carefully explained review. I was particularly impressed that the authors followed up their review by outlining current research into combating this resistance.
I have very little to add to these comments except some very minor typos and observations on the text.
I recommend that this publication is accepted for publication by the journal after some very minor typo corrections.
Dear reviewer,
Thank you for your detailed and constructive feedback on our manuscript titled "Colistin Resistance Mechanism and Management Strategies of Colistin-Resistant Acinetobacter baumannii Infections" We appreciate the time and effort you have invested in reviewing our work and providing valuable insights. We have addressed each of your comments and suggestions to improve the manuscript.
Minor comments
Line 19. Perhaps replace the words Bugs in the abstract with something more formal since it is an in-depth review.
Reply: Thank you for this insightful comment. We replaced “bugs’ with “pathogen”.
Line 37. Should A. baumanii not be in italics? Be careful of the spacing when you abbreviate.
Reply: Thank you for this observation. In the revised manuscript we have italicized all species names until the family level.
Line 96. Confer?
Reply: Thank you. It will be a “confer”. In the revised manuscript we change it.
Line 106. Is this reference the same as line 111?
Reply: Thank you for this comment. In lines 106 and 111 the references are different.
Figure 2. Nice diagram.
Reply: Thank you for your appreciation!
Line 174. Not sure about starting a sentence with an abbreviated gene name?
Reply: Thank you for this comment. Insertion sequences of A. baumanni are generally termed as ISAba. In the revised manuscript, we rectify it.
Line 509. The authors are correct, extreme caution should be exercised using CRISPR methods to avoid unintended genetic alterations.
Reply: Thank you for your appreciation!

Reviewer 3 Report
Comments and Suggestions for Authors
This is a review article so authors have given the Figure 1. Mechanisms of action of colistin against A. baumannii. SOD (Superoxide dismutase). (Cre- 125 ated with Biorender.com) based on literature so please provide the reference for this literature.
Same for Figure 2. A schematic illustration showing the general mechanisms of colistin resistance in A. bau- 151 mannii. LPS (Lipopolysaccharide), GalN (Galactosamine), PEtN ( Phosphoethanolamine). (Created 152 with Biorender.com).
Overexpression is Figure 2 is not clear and confusing for readers.
It will be much better if 3. Mechanism of Colistin resistance in A. baumannii can be divided in sub-sections.
Conclusion section should be revised to make it really conclusive and remove the statements that still need references.
Comments on the Quality of English LanguageTypo errors can be removed by extensive reading.
Author Response
Dear Reviewer,
Thank you for your detailed and constructive feedback on our manuscript titled "Colistin Resistance Mechanism and Management Strategies of Colistin-Resistant Acinetobacter baumannii Infections" We appreciate the time and effort you have invested in reviewing our work and providing valuable insights. We have addressed each of your comments and suggestions to improve the manuscript. Below, we outline the revisions made in response to your feedback:
This is a review article so authors have given the Figure 1. Mechanisms of action of colistin against A. baumannii. SOD (Superoxide dismutase). (Cre- 125 ated with Biorender.com) based on literature so please provide the reference for this literature.
Reply: Thank you for this valuable comment. We modified the figure description with references.
Same for Figure 2. A schematic illustration showing the general mechanisms of colistin resistance in A. bau- 151 mannii. LPS (Lipopolysaccharide), GalN (Galactosamine), PEtN (Phosphoethanolamine). (Created 152 with Biorender.com).
Reply: Thank you so much. We modified the figure accordingly and clarified the description to avoid any ambiguity.
Overexpression is Figure 2 is not clear and confusing for readers.
Reply: Thank you for this insightful observation. We modified the figure and separated each mechanism to remove confusion.
It will be much better if 3. The mechanism of Colistin resistance in A. baumannii can be divided in sub-sections.
Reply: Thank you for your valuable suggestion. In the revised manuscript we divided the “Mechanism of Colistin resistance in A. baumannii” into three subsections.
Conclusion section should be revised to make it really conclusive and remove the statements that still need references.
Reply: Thank you. We concise the conclusion section according to your valuable suggestion.

Round 2
Reviewer 1 Report
Comments and Suggestions for Authors
The manuscript was improved and could be accepted for publication
Comments on the Quality of English LanguageMinor spelling errors could be corrected